# Mangrove Species Classification from Unmanned Aerial Vehicle Hyperspectral Images Using Object-Oriented Methods Based on Feature Combination and Optimization

**DOI:** 10.3390/s24134108

**Published:** 2024-06-24

**Authors:** Fankai Ye, Baoping Zhou

**Affiliations:** College of Information Engineering, Tarim University, Alaer 843300, China; 15024688199@163.com

**Keywords:** mangrove, feature combination, object-oriented, unmanned aerial vehicles, hyperspectral image classification

## Abstract

Accurate and timely acquisition of the spatial distribution of mangrove species is essential for conserving ecological diversity. Hyperspectral imaging sensors are recognized as effective tools for monitoring mangroves. However, the spatial complexity of mangrove forests and the spectral redundancy of hyperspectral images pose challenges to fine classification. Moreover, finely classifying mangrove species using only spectral information is difficult due to spectral similarities among species. To address these issues, this study proposes an object-oriented multi-feature combination method for fine classification. Specifically, hyperspectral images were segmented using multi-scale segmentation techniques to obtain different species of objects. Then, a variety of features were extracted, including spectral, vegetation indices, fractional order differential, texture, and geometric features, and a genetic algorithm was used for feature selection. Additionally, ten feature combination schemes were designed to compare the effects on mangrove species classification. In terms of classification algorithms, the classification capabilities of four machine learning classifiers were evaluated, including K-nearest neighbor (KNN), support vector machines (SVM), random forests (RF), and artificial neural networks (ANN) methods. The results indicate that SVM based on texture features achieved the highest classification accuracy among single-feature variables, with an overall accuracy of 97.04%. Among feature combination variables, ANN based on raw spectra, first-order differential spectra, texture features, vegetation indices, and geometric features achieved the highest classification accuracy, with an overall accuracy of 98.03%. Texture features and fractional order differentiation are identified as important variables, while vegetation index and geometric features can further improve classification accuracy. Object-based classification, compared to pixel-based classification, can avoid the salt-and-pepper phenomenon and significantly enhance the accuracy and efficiency of mangrove species classification. Overall, the multi-feature combination method and object-based classification strategy proposed in this study provide strong technical support for the fine classification of mangrove species and are expected to play an important role in mangrove restoration and management.

## 1. Introduction

Mangrove forests rank among the most productive ecosystems on Earth and act as natural barriers to tropical and subtropical coastlines [1,2]. Mangrove ecosystems play a critical role in seawater purification, wind and wave prevention, biodiversity maintenance, and carbon sequestration and storage [3,4,5], providing vital ecosystem services for coastal areas [6]. Unlike other forest ecosystems, mangrove ecosystems are notably fragile and susceptible to human activities and climate change [7]. Over recent decades, global mangrove forests have experienced significant degradation [8,9], with a reported decrease of approximately 35% between 1980 and 2005 [10,11,12]. Hence, precise monitoring of mangrove forest species composition and spatial distribution is essential for effective management and conservation efforts.

The species composition and distribution of mangrove forests are crucial for the restoration and protection of mangrove ecosystems [13]. Traditionally, species surveys of mangrove forests have depended on regular manual surveys for data collection [14]. While manual surveys yield precise data, particularly beneficial for mangrove ecosystem restoration and conservation, they are generally limited to small spatial scales. Moreover, manual surveys incur higher costs in terms of time and expenses, especially in survey areas encompassing tidal, muddy, and remote coastal ecosystems.

In recent years, with the technological advancement of optical sensors, remote sensing has become increasingly prevalent in grassland resource surveys [15], precision agriculture [16], and target detection [17]. Similarly, in mangrove ecosystems, the utilization of remote sensing techniques for species surveys of mangrove forests has seen a gradual increase [18], owing to their discernible spatial characteristics and visible spectra on remote sensing images. Satellite remote sensing images, capable of capturing vast areas at once, have been employed by scholars to monitor mangrove ecosystems. For example, Ghorbanian et al. [19] utilized Sentinel satellite images to attempt the identification of mangrove forests, while Lee et al. [20] employed Landsat satellite data to monitor the degradation extent of mangrove ecosystems. Nonetheless, the limited resolution of satellite remote sensing systems often hampers the accurate depiction of subtle changes and species distinctions within mangrove forests [21], thereby constraining the efficacy of satellite imagery in mangrove forest recognition. Moreover, the presence of cloud cover diminishes the imaging quality of satellite images. Recently, unmanned aerial vehicle (UAV) remote sensing has emerged as a promising technological solution to mitigate the drawbacks of satellite remote sensing [22]. The agility and maneuverability of UAVs facilitate the acquisition of more time-series data, thereby enhancing the monitoring precision and depth of comprehension of mangrove ecosystems [23]. Onishi et al. [24] utilized a UAV to capture red–green–blue (RGB) images and segmented canopy objects based on color and three-dimensional information to achieve the identification of seven tree species. Similarly, Santos et al. [25] employed RGB images and a target detection algorithm to investigate protected tree species, while Franklin et al. [26] utilized multispectral images for nine tree species. Compared to images captured by other optical sensors (RGB and multispectral), hyperspectral images can capture more intricate object details due to containing hundreds or even thousands of continuous bands [27]. Thus, the integration of UAVs with hyperspectral sensors enables finer mangrove species surveys and ecological monitoring [28]. Qin et al. [29] conducted tree species classification utilizing hyperspectral and RGB images, with results indicating higher classification accuracy for hyperspectral images. The development of UAV hyperspectral remote sensing platforms provides a robust monitoring tool for the ecological survey of mangrove forests.

Currently, the task of species classification in mangrove forests can be categorized into pixel-based and object-based processing methods depending on the image processing approach [30,31]. Pixel-based image processing methods are generally more intuitive and easier to comprehend, and they tend to perform reliably in classifying objects with significant spectral differences. However, this approach is susceptible to image quality and noise, which may lead to errors [32]. Moreover, the occurrence of the salt-and-pepper phenomenon in classification results has been reported [33]. In contrast, object-based image processing methods can address the limitations of pixel-based methods and yield classification results with greater clarity and density [34]. Recently, machine learning and object-oriented classification methods have emerged for mangrove classification. Zhang et al. [35] demonstrated that object- and random forest-based classification methods achieve higher mapping accuracy in mangrove mapping tasks. Similarly, Zhou et al. [36] emphasized the effectiveness of object and machine-learning-based classification methods in enhancing classification accuracy. However, due to the considerable variability and spatial heterogeneity of mangrove forests, species classification remains challenging [37]. To address this challenge, Ou et al. [38] considered the texture structure of tree species and improved mangrove species classification accuracy by incorporating texture features. Additionally, Cao et al. [39] showed that first-order differential methods could enhance inter-species differences and improve the classification accuracy of mangrove species. Despite the promising classification accuracy achieved by the aforementioned methods, few studies have comprehensively assessed the impact of combining multiple feature variables on the classification performance of mangrove forests in hyperspectral images. Furthermore, object-oriented feature combinations and optimization based on object-oriented features have not been fully explored.

To address the aforementioned issues, we adopt an object-based approach to evaluate the influence of different combinations of feature variables and feature selection on mangrove species classification. Therefore, the objectives of this paper are as follows: (1) To investigate the effect of individual feature variables on the classification accuracy of mangrove tree species and identify the significant subset of features within each variable; (2) to compare the impact of different combinations of feature variables on the classification accuracy of mangrove tree species; and (3) to assess the effect of feature selection on the classification accuracy of machine learning algorithms, and to compare the performance of four machine learning algorithms—the K-nearest neighbor (KNN), support vector machine (SVM), random forest (RF), and artificial neural network (ANN)—to determine the optimal classification model. This study aims to present a more precise and efficient technical methodology for mangrove classification, thereby providing robust technical support for the monitoring and management of mangrove ecological resources.

## 2. Materials and Methods

### 2.1. Overview of the Study Area

Futian Mangrove Nature Reserve is situated in the northeastern part of Shenzhen Bay (22°31′36″ N, 114°0′30″ E), encompassing an area of 368 hectares, making it the smallest national nature reserve in China. This area falls within the East Asian monsoon zone and the South Asian tropical maritime monsoon climate zone. The mean annual temperature is 22.4 °C, with a monthly average of 14.1 °C. The lowest temperature is recorded in January, while the highest occurs in July. The highest temperature on record is 38.7 °C, and the lowest is 0.2 °C. Annual rainfall averages between 1700–1900 mm, predominantly occurring from April to September. The reserve comprises a core area, a buffer area, and an experimental area. The buffer zone serves as a transition area from wetland to land, boasting rich and diverse species. Key community species include *Kandelia obovata*, *Avicennia marina*, *Sonneratia caseolaris*, and *Sonneratia apetala*, etc. Therefore, for this study, mangrove forests within the buffer zone were primarily selected for UAV hyperspectral remote sensing data collection (Figure 1).

### 2.2. UAV Hyperspectral Image Data

The high-resolution hyperspectral image data were acquired in November 2017, between 10:00 and 15:00 Beijing time, under clear sky conditions with sparse clouds and minimal ground-level wind disturbance. The data were captured using an XC-1300 UAV fitted with a ZK-VNIR-FPG480 (ZKYD Data Technology Co., Ltd., Beijing, China) push-sweep hyperspectral sensor, operating at an altitude of 100 m and offering a spatial resolution of 40 cm. The UAV flight path ensured a heading overlap of 60% and a side-to-side overlap of up to 30%. The hyperspectral image sensor covers the spectral range of 400–1000 nm, with a spectral resolution of 2.8 nm, and comprises 270 spectral bands. However, for this study, bands within the ranges of 412.6–1011.4 nm and 430.4–991.3 nm were excluded due to significant noise interference at both ends, resulting in a final selection of 253 bands for analysis.

### 2.3. Field Survey and Sample Collection

Field surveys were conducted in January and November 2017 at Futian Mangrove Nature Reserve in Shenzhen, where the study site predominantly comprised four mangrove species: *Kandelia obovata*, *Avicennia marina*, *Sonneratia caseolaris*, and *Sonneratia apetala*. Table 1 outlines the characteristics of these mangrove species alongside the localized views on the UAV reference images. A survey sample plot size of 5 m × 5 m was established, with a total of 86 sample points surveyed for each species. Recorded data included the geographic coordinates of each sample site and species photographs to assist in sample selection. Representative object samples for each category were chosen based on multi-scale segmentation techniques, as described in Section 2.7. Specifically, we selected samples of segmented objects in ArcGIS 10.7 software and obtained spectral samples by the average value of each object. These samples are evenly spaced at different locations in the study area. Overall, 7045 object samples were gathered for mangrove species classification, with 2651 samples for *Kandelia obovata*, 1203 for *Avicennia marina*, 1565 for *Sonneratia caseolaris*, and 1626 for *Sonneratia apetala*. Thirty percent of these samples were randomly allocated for model training, while the remainder were reserved for validation (Table 2).

### 2.4. Spectral Transformation

Fractional order differentiation of spectra offers a means to mitigate the impact of light variations and to enhance spectral discrepancies between different species, making it a commonly employed technique in hyperspectral vegetation studies [40]. Among the fractional order differentiation methods, Caputo, Grünwald–Letnikov, and Riemann–Liouville approaches are frequently utilized [41]. Given its simplicity and suitability for spectral signal processing [42], the Grünwald–Letnikov approach was selected for extracting feature information related to the first-order differentiation (FOD) and second-order differentiation (SOD) of the mangrove reflectance spectra in this investigation. Grünwald–Letnikov’s formula is defined as follows:(1)$$d^{\nu }f\left(x\right)=\underset{h\to 0}{lim}\frac{1}{h^{\nu }}\sum_{m=0}^{\left[\left(b-a\right)/h\right]}(-1{)}^{m}\frac{\Gamma \left(\nu +1\right)}{m!\Gamma \left(\nu -m+1\right)}f\left(x-mh\right),$$
where *v* > 0 denotes the differential order, *x* represents the spectral reflectance value, *h* denotes the differential step, *b* and *a* denote the upper and lower limits of the differentiation, respectively, and Γ represents the Gamma function. The formula is as follows:(2)$$\Gamma \left(z\right)=\int_{0}^{\infty }exp\left(-u\right)u^{z-1}du=\left(z-1\right)!,$$

### 2.5. Vegetation Index

Vegetation indices (VIs) are typically mathematical transformations of spectral reflectance designed to reveal hidden vegetation information. VIs achieve this by scaling, differencing, and applying linear combinations to enhance the differences between plant species [43]. Through a review of pertinent literature [44,45,46], 15 hyperspectral VIs that are instrumental in classifying mangrove forests were identified, as presented in Table 3.

### 2.6. Texture Features

Texture features (TFs) are characteristics that describe surface details and structures present in an image, independent of image brightness [47]. In the field of computer vision, texture features find widespread usage in tasks, such as target classification and image segmentation [48]. Given the diverse species of mangrove forests with distinct texture features, the Gray Level Co-occurrence Matrix (GLCM) is selected to capture texture variations in the spatial direction of tree species. Due to the high correlation among numerous bands in hyperspectral images, the principal component analysis (PCA) algorithm is employed to extract the first three principal component bands for GLCM calculation. It is worth noting that these three principal component bands retain 99.04% of the features from the original image. Based on these three principal component bands, a total of 8 texture features (mean, variance, homogeneity, contrast, dissimilarity, entropy, second moment, and correlation) are extracted, amounting to 24 feature variables. Furthermore, considering the influence of different window sizes on the extraction of texture features, a total of 22 window sizes ranging from 3 × 3 to 45 × 45 (odd-sized) are implemented.

### 2.7. Multi-Scale Segmentation

Multi-scale segmentation involves grouping neighboring pixels with similar attributes into a single object, with the image object serving as the unit for object-oriented classification [49]. In the context of mangrove forests, multi-scale segmentation proves effective in extracting tree objects from homogeneous regions due to the significant variations in tree structures. Furthermore, multi-scale segmentation reduces the number of samples to be classified in the image, thereby enhancing monitoring efficiency. The segmentation scale represents a crucial parameter in object-oriented analysis [50]. In this study, multi-scale segmentation was performed using the ENVI 5.3 software, and optimal segmentation parameters were chosen by setting the segmentation scale and merging scale. The impact of different segmentation parameters on mangrove segmentation is illustrated in Figure 2. From the figure, it is evident that smaller segmentation scales result in denser segmentation effects, while larger segmentation scales exhibit greater sensitivity to changes in brightness. Increasing the merging scale leads to the merging of tree species with the same attribute, but excessive merging may result in the loss of valuable information due to variations in lighting conditions. Considering these factors, the segmentation parameter with a segmentation scale of 0 and a merging scale of 80 was selected for extracting the mangrove objects. In total, 14 geometric features (GFs) of the extracted objects were utilized in this study, including area, length, compactness, convexity, solidity, roundness, form factor, elongation, rectangularity fit, main direction, major axis, minor axis, number of holes, and solid area.

### 2.8. Research Methods and Accuracy Assessment

#### 2.8.1. Classification Methods

In this study, a total of four classification methods were selected, including KNN, SVM, RF, and ANN.

The KNN algorithm is a distance-based classification method that does not require an explicit training process [51]. When constructing the KNN model, classification entails identifying the K-nearest neighbors to the sample and subsequently classifying these neighbors based on their labels. In this study, the parameter K = 17 for KNN was determined through a grid search.

The RF is a composite learning method comprised of multiple decision trees [52]. The RF makes collective decisions by aggregating the classification outcomes of numerous decision trees, thereby yielding relatively accurate predictions. During training, each decision tree employs a random subset of features, mitigating the risk of overfitting and enhancing the model’s generalization capability. The quantity of decision trees serves as a crucial parameter for RF classifiers, with the optimal number determined to be 500 through a grid search.

The SVM is a machine learning algorithm primarily used for binary classification. Its learning strategy aims to identify the hyperplane that maximizes the interval in the feature space [53]. The SVM classifier relies on two important parameters: the penalty coefficient, C, and the kernel function. The kernel function plays a crucial role in mapping low-dimensional nonlinear data to a higher-dimensional feature space, enabling the transformation of linearly indivisible data into linearly divisible data in the new space. For this study, we performed a grid search and found the optimal value for C to be 200. Additionally, we selected the radial basis function (RBF) as the kernel function.

The ANN is a computational model inspired by biological neural systems [54]. It comprises numerous interconnected artificial neurons organized into input, hidden, and output layers. Throughout the training process, the classification ability of the network is adjusted by constantly updating the weights between neurons. For this study, we employed a grid search to determine the parameters of the ANN. Specifically, we set the number of neurons to 64, used the sigmoid activation function, employed the Adam optimization function, and conducted 100 iterations.

All the above algorithms were implemented in the Python 3.6.5 programming language by using the sklearn library.

#### 2.8.2. Accuracy Assessment

To assess the performance of different models, we selected several evaluation metrics based on the confusion matrix, namely overall accuracy (OA), average accuracy (AA), and Kappa coefficient. OA represents the percentage of correctly classified samples out of the total number of samples. AA represents the average accuracy across all categories. The Kappa coefficient measured the degree of agreement between the classified results and the actual samples. Additionally, to provide reliable evaluation results, the 10-fold cross-validation technique was used on the training set.

## 3. Results

### 3.1. Spectral Feature Analysis

The curves of the original spectrum (OS) and fractional order differentials for different mangrove species are depicted in Figure 3. Observing Figure 3a, it becomes apparent that the spectral curves of different tree species exhibit similar trend variations. Notably, the spectral curves of *Kandelia obovata* and *Avicennia marina* show greater similarity, while those of *Sonneratia caseolaris* and *Sonneratia apetala* exhibit a similar pattern, possibly due to their similar textures and color characteristics (Table 1). Fractional order differentials can be used to describe the instantaneous change in a function at a given point. As shown in Figure 3b, following the FOD of the OS, the characteristic difference between the features of *Kandelia obovata* and *Avicennia marina* at 722 nm was accentuated, thereby enhancing their distinguishability. In contrast, the spectral curves of SOD exhibited relative stability without excessive peaks, tending to stabilize around 0. In summary, fractional order differential contributes significantly to amplifying spectral differences among similar species, which is crucial for accurate tree species classification.

### 3.2. Optimal Texture Window Sizes

TFs are recognized as crucial parameters for mangrove extraction. TF values obtained from varying window sizes exhibit variance. Given that tree diameters are relatively large compared to herbaceous plants, smaller window sizes may not be conducive to identifying mangrove tree species effectively. In this section, we assess the impact of different texture window sizes on tree species identification by utilizing the Jeffries–Matusita (J-M) distance as a measure of separability between tree species classes under various window sizes (see Figure 4), where higher values indicate greater separability. From the figure, it is apparent that both *Kandelia obovata* and *Avicennia marina* demonstrate optimal separability across different window sizes, while smaller window sizes result in diminished separability for other species. Specifically, as the window size increases, the separability between tree species gradually improves. Notably, at a window size of 23, the J-M distance between all tree species stabilizes without significant alteration. Consequently, a window size of 23 is deemed optimal for extracting texture features.

### 3.3. Optimal Feature Selection

A plethora of feature variables can introduce redundant features, consequently diminishing model performance. Thus, feature selection is deemed an effective method for enhancing model performance by reducing model complexity and feature correlation. The genetic algorithm, a heuristic optimization algorithm, mimics genetic and fitness selection mechanisms observed in biological evolution. In this study, a genetic algorithm is employed to select optimal features from various feature variables. Specifically, we utilize different gene sizes to select the most crucial features, denoting selected features as 1 and unselected features as 0. Within the genetic algorithm framework, we set the population size to 50, the crossover probability to 0.9, the mutation probability to 0.2, the gene size to correspond to the number of features, and the number of iterations to 20. The final selection of feature numbers in this paper is adaptively chosen through a genetic algorithm. The genetic algorithm is implemented in Python, and through continuous iteration and optimization, it can accurately select the important feature parameters for each model to improve model accuracy. The result of the optimal features selected by the genetic algorithm is depicted in Figure 5. The numbers in brackets represent the number of initial and selected features.

The number of features after the optimization has been significantly reduced, as can be seen in Figure 5. For the original spectral (OS) curves, the features selected by the four models are primarily concentrated in the spectral ranges of 508–575 nm and 708–744 nm, consistent with the results in Figure 3a. In the case of FOD, the selected features are concentrated in the spectral ranges of 497–552 nm and 706–717 nm. Meanwhile, for SOD, the selected features are focused on the spectral ranges of 528–575 nm and 706–759 nm. Overall, features selected multiple times in the OS and spectral transformations are primarily concentrated in two regions: the green peak and the near-infrared band. The selected bands were primarily chosen for their significance in the spectral response of plants and their effects on plant physiology and biochemistry. Specifically, the green-peak band (500–600 nm) and the near-infrared band (700–800 nm) were widely selected. The green-peak band is indicative of the photosynthetic activity of chlorophyll, while the near-infrared band reflects changes in plant cellular structure and water content. Regarding TFs, the importance of texture features derived from different principal components varies. However, mean, variance, and contrast were selected by multiple models, indicating their significance in mangrove species classification. Conversely, entropy contributed less to mangrove species classification in the feature selection results. Among the characteristic variables of the vegetation index, BGI2, TCARI, TCARIOSAVI, PRI, CRI2, and NLI made substantial contributions to mangrove species classification as they were selected by four models. Among GF variables, area, convexity, major axis, and solid area exhibited significant contributions, while elongation had the smallest contribution. In summary, feature selection effectively reduces data redundancy and enhances the classification accuracy of the models.

### 3.4. Classification of Single Feature Variables

To evaluate the significance of various feature variables for mangrove species classification, we experimented with a single feature variable. Additionally, to assess the effectiveness of the optimal features, we compared them with the unselected feature variables, and the results are presented in Table 4. Please note that only the accuracy results of OA are provided in the table.

As shown in Table 4, the classification accuracy improves significantly after selecting the features from different feature variables. These results highlight the importance of appropriate feature selection in reducing dimensionality and enhancing model performance. Moreover, when comparing the performance of four machine learning algorithms, SVM and ANN outperform the others in extracting mangrove information, achieving the highest classification accuracy. Specifically, SVM demonstrates exceptional performance when using a single feature variable, effectively equalizing the accuracy across all feature variables. Among the feature variables, TF based on the SVM model attains the highest accuracy (97.04%), underscoring its crucial role in classifying mangrove species. Following closely are FOD and SOD based on the ANN model, both surpassing 96% classification accuracy. On the other hand, OS exhibits relatively lower accuracy, with the highest classification accuracy achieved at 95.72% using the SVM model. Notably, the classification accuracy of GF is quite poor, with the highest accuracy reaching only 39.33%, thus indicating its inability to solely recognize different mangrove species.

### 3.5. Classification of Combined Feature Variables

According to the experimental results in Table 4, SVM demonstrates good applicability for multiple feature variables, achieving an average accuracy of 85.19% for all variables, which is higher than that of the other three models. Therefore, we chose SVM for the screening of combined feature variables. To explore the impact of various image feature combinations on mangrove species classification accuracy, this study designed 10 classification schemes, with the experimental findings detailed in Table 5.

From Table 5, it is evident that the accuracy of Scheme 1 decreases to 95.68% after introducing FOD into OS, owing to FOD and OS sharing the same spectral feature. This redundancy weakens the model’s classification performance. Scheme 2 reveals that incorporating FOD into TF reduces the model’s accuracy. However, Scheme 3, combining FOD and TF in OS, enhances the model’s classification performance, underscoring the significance of FOD and TF. Moreover, in Scheme 4, FOD proves more effective in extracting feature differences in OS compared to SOD. Introducing VI in Scheme 5 further boosts classification accuracy, highlighting VI’s importance as a feature variable. Similarly, introducing GF in Scheme 6 improves accuracy, albeit to a slightly lower degree than in Scheme 5. Notably, Schemes 7 and 8 demonstrate VI and GF’s comparable importance, both achieving an OA value of 97.22%. Scheme 9, combining VI and GF, significantly enhances classification accuracy, suggesting their complementary nature. Additionally, Scheme 10, combining the remaining feature variables except SOD, achieves the highest classification accuracy, further emphasizing the importance of combining feature variables in mangrove species classification.

Overall, the classification accuracies varied across different combinations of feature variables. The contributions of VI and GF to mangrove classification were nearly equal, with VI showing a slightly higher contribution than GF, consistent with the findings in Table 4. FOD and TF played significant roles in mangrove species classification, warranting priority consideration in feature extraction. While VI and GF made comparatively weaker contributions, they serve as auxiliary feature variables, enhancing the model’s classification performance.

### 3.6. Comparison of Different Classification Methods

The results comparing the ability of different classification models to classify mangrove species based on the optimal combination of feature variables are presented in Table 6. From the table, it is evident that the ANN model, utilizing the optimal feature variable combination, achieves the highest classification accuracy, with OA, AA, and Kappa values of 98.03%, 97.93%, and 0.9729, respectively. This superiority can be attributed to the superior nonlinear processing capability of ANN and its adeptness at learning features in high-dimensional data. SVM, while slightly trailing the ANN model, still achieves a commendable OA value of 97.59%. Overall, both SVM and ANN models demonstrate accurate mangrove species recognition, with individual species recognition accuracies exceeding 97%. Conversely, KNN exhibits the poorest classification performance, primarily due to increased feature dimensions limiting its ability to improve classification accuracy. Furthermore, KNN is an instance-based nonparametric method that relies on calculating the distance between each test sample and all training samples. As a result, the distance calculation becomes inaccurate in high-dimensional data, leading to a degradation in classification performance. In contrast, RF attains an overall classification accuracy of 94.28%, outperforming KNN. This improvement can be attributed to RF’s integrated nature, which mitigates the impact of high-dimensional features by aggregating decisions from multiple decision trees. Additionally, the cross-validation results show that the cross-validation accuracy of all models does not significantly differ from the accuracy on the test set, indicating good model stability and no overfitting. For a more detailed analysis of generalization capability, please refer to Section 4.3. The accuracy of the method is further validated by the confusion matrix results in Figure 6.

### 3.7. Visualization of Classification Results

To facilitate a clear and intuitive comparison of different methods in classifying mangrove species, Figure 7 illustrates the classification result maps and local zoom maps of various methods. From the figure, it is apparent that KNN exhibits relatively poor recognition, with numerous misidentified samples and frequent confusion between *Sonneratia caseolaris* and *Avicennia marina*. In contrast, the classification result of RF surpasses that of KNN, featuring clearer boundaries between different species and fewer instances of confusion, notably between *Sonneratia caseolaris* and *Avicennia marina*. Both SVM and ANN demonstrate the best classification results, effectively recognizing different species, with their relative aggregation aligning with actual distribution. Compared to SVM, ANN displays an advantage in identifying species at the boundary, mitigating instances where *Kandelia obovata* and *Kandelia marina* are incorrectly classified as the same species.

## 4. Discussion

### 4.1. Effect of Texture Window on Classification Results

Texture features are adept at capturing spatial gray-scale information and the structural characteristics of an image. Given the high spectral similarity of mangroves (Figure 3a), achieving accurate mangrove species recognition solely through spectra is challenging. However, different species exhibit distinct texture structures, offering the potential for the precise identification of mangrove species. This notion is further supported by the results in Table 4, which underscore the importance of texture features. The window size plays a crucial role in extracting texture features, with varying accuracies obtained using different window sizes [55]. Figure 8 displays a line graph depicting the percentage improvement in accuracy with different window sizes. The graph reveals that the recognition accuracy of mangroves stabilizes as the window size increases. When the window size reaches 23, the improvement in mangrove recognition accuracy becomes insignificant, consistent with the findings in Figure 4. The appropriate window size depends on the size of the target being recognized. Xiao et al. [56] emphasized that smaller windows are suitable for smaller objects, whereas larger windows are more suitable for larger objects. Given the relatively substantial structure of mangrove forests, smaller windows yield lower accuracy. Conversely, the selected window size in this study aligns with the actual size of mangrove forests, resulting in higher accuracy. Saboori et al. [57] found that a window size of 51 × 51 is most effective for extracting urban land use, and Duan et al. [58] discovered that window sizes of 19 × 19 and 23 × 23 improve soil identification accuracy. These findings align with the observations in this paper, where the appropriate window size enhances recognition accuracy by considering the variation in object sizes.

### 4.2. Comparison of Object-Based and Pixel-Based Classification

To compare the advantages and disadvantages of object-based and pixel-based classification methods in mangrove species identification, this study conducts a comparative experimental analysis. The findings are presented in Table 7. Additionally, for intuitive observation of the classification outcomes, the results are visualized using the ANN model with the highest classification accuracy, as depicted in Figure 9.

Table 7 illustrates that the ANN model performs optimally in both object-based and pixel-based classification, consistent with the findings from Table 6. Regarding classification accuracy, the performance of both the object- and pixel-based models, in descending order, ranks as ANN > SVM > RF > KNN. In pixel-based classification, the ANN achieves the highest accuracy, with an OA of 96.86%, an AA of 96.79%, and a Kappa coefficient of 0.9572. Across all categories, the ANN model achieves a classification accuracy exceeding 95%. Object-based classification exhibits superior performance compared to pixel-based classification, with a 1.14% increase in OA, a 1.14% improvement in AA, and a 0.0157 increase in the Kappa coefficient. Generally, object-based methods prove more suitable for mangrove species classification than pixel-based methods. This superiority stems from the ability to classify pixels within homogeneous regions as having the same attribute through multi-scale segmentation, thereby mitigating spectral variations due to lighting and color discrepancies and consequently enhancing recognition accuracy within homogeneous regions. Luo et al. [59] similarly support this conclusion.

Based on the classification results depicted in Figure 9, it is evident that both object-based and pixel-based methods accurately extract the spatial distribution of different mangrove species. However, upon closer examination of the localized zoomed-in image, it becomes apparent that the pixel-based method exhibits a significant amount of salt-and-pepper noise. In contrast, the object-based method effectively avoids this salt-and-pepper phenomenon. Zhu et al. [60] also made a similar observation, noting that the object-oriented method accurately extracts leaf vein networks while minimizing the presence of salt-and-pepper noise. Although the object-based method only yields a modest 1% improvement in accuracy, it demonstrates impressive performance in terms of the visual classification results.

### 4.3. Impact of Sample Numbers

To verify the generalization ability of the classification strategy proposed in this paper, we selected different training samples for each category, and the results are shown in Figure 10. The figure indicates that the accuracy of the model gradually improves with the increase in sample size. Among the models, KNN performs the worst with varying numbers of training samples due to the original features’ poor separability, leading to suboptimal classification results when using the distance metric. Compared to KNN, RF achieves better classification accuracy because it enhances model robustness by integrating multiple decision trees and making decisions through voting. SVM effectively improves classification accuracy by transforming the data through a kernel function. In contrast, ANN excels in handling nonlinearity and, through continuous learning, can identify subtle differences in the features of different tree species, thereby enhancing the model’s classification performance. In experiments with different training samples, ANN consistently performs better and maintains the highest classification accuracy, further demonstrating its classification advantage and good generalization ability under multi-feature combinations.

### 4.4. Running Time Analysis

Table 8 shows the running times for different models. From the table, it can be seen that RF has the longest training time due to the large number of decision trees, which increases the training duration. KNN has the shortest training time because it is a nonparametric model, thereby reducing training time costs. SVM has a training time of only 0.153 s, but its prediction time is the longest, likely because SVM needs to calculate the distance between the samples and all the support vectors during prediction, increasing the prediction time. ANN has a training time of 2 s, which is second only to RF, but its prediction time is the shortest, resulting in higher prediction efficiency. ANN learns to adapt to the task of classifying mangrove species through continuous iteration during training, which increases the training time. Generally, training time increases with the number of iterations. However, once training is complete, the model complexity is lower due to ANN’s fewer parameters, which improves prediction efficiency. Overall, the ANN model demonstrates better classification efficiency and accuracy in mangrove classification.

### 4.5. Limitations and Outlook

A species survey of mangrove forests serves as a crucial foundation for the protection of these ecosystems. In this study, species classification of the mangrove forests was achieved through feature selection and combination using an object-oriented approach, but there are some limitations. Firstly, the determination of the optimal segmentation scale in the object-oriented method relied on a step-by-step trial-and-error process, which was time-consuming and tedious, although it yielded improved segmentation results. Future research should explore the development of adaptive segmentation methods specifically designed for object-oriented analysis. Secondly, while this study achieved favorable classification accuracy through feature combination, it unavoidably increased the dimensionality of the data. Therefore, it is essential to investigate the impact of various feature combinations on classification accuracy in future studies and to devise strategies to address the dimensionality problem arising from feature combinations. Lastly, the classification method employed in this study is an example of machine learning, which relies on expert knowledge for feature extraction. Deep learning, as an end-to-end learning approach, could be considered for feature extraction and integrated with object-oriented methods to further enhance classification efficiency. Nonetheless, this study still holds advantages in the species classification task of mangrove forests and can provide valuable technical support for the ecological protection and management of these vital ecosystems.

## 5. Conclusions

This paper presents a study on mangrove species classification utilizing hyperspectral image data collected by UAVs in the Futian Mangrove Nature Reserve in Shenzhen. Through the integration of an object-oriented approach with feature combination and optimization techniques, we established an optimal model specifically designed for accurate mangrove classification. Extensive experimental validation was conducted to confirm the effectiveness of our proposed method. The key findings of this research are summarized as follows:

(1) In this paper, we employ a genetic algorithm for feature selection. Our findings demonstrate the necessity of feature selection in high-dimensional datasets. This process effectively reduces the dimensionality of feature variables while improving the performance of classification models.

(2) The effectiveness of individual feature variables for classifying mangrove species was assessed. The analysis revealed that both texture features and differential spectra play crucial roles in mangrove species classification. Specifically, texture features utilized with SVM achieved an OA of 97.04%, whereas ANN applied to second-order differential spectra achieved an OA of 96.47%. In addition, experiments with geometric features yielded the lowest effectiveness among the single feature variables, with an overall classification accuracy of below 40%.

(3) The study compared the impacts of different feature combinations on classifying mangrove species. The results show that the ANN, using a combination of five feature variables, achieved the highest classification performance, with OA, AA, and Kappa values of 98.03%, 97.93%, and 0.9729, respectively. Additionally, the vegetation index and geometric features can act as supplementary features to improve the model’s classification accuracy.

(4) Four classification models, namely KNN, SVM, RF, and ANN, were compared in terms of their classification accuracies, which were ranked from high to low as follows: ANN > SVM > RF > KNN. Furthermore, this study also compared the performance of object-based and pixel-based classification methods. The findings revealed that the object-based method achieved a higher classification accuracy while successfully avoiding the salt-and-pepper phenomenon.

## Figures and Tables

**Figure 1 sensors-24-04108-f001:**
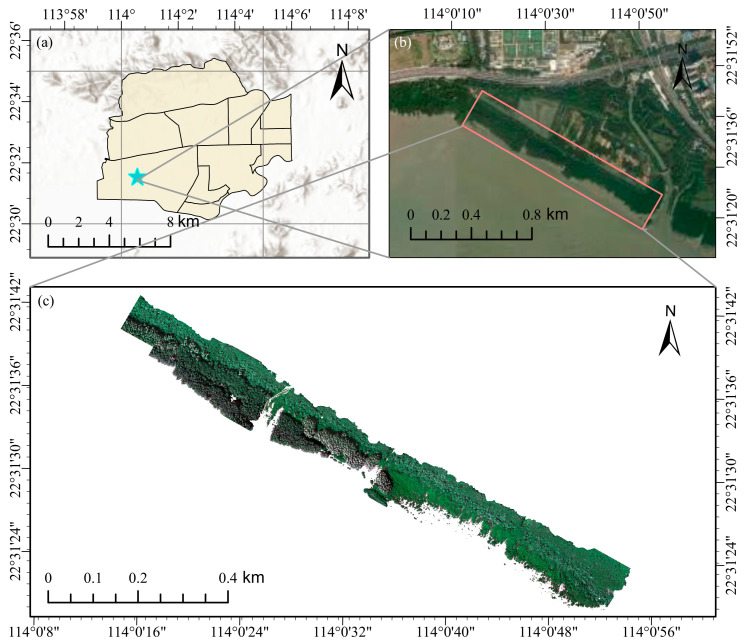
Study area and RGB image. (**a**) Geographic location of the study area. (**b**) Satellite high-resolution image of the study area. (**c**) RGB image.

**Figure 2 sensors-24-04108-f002:**
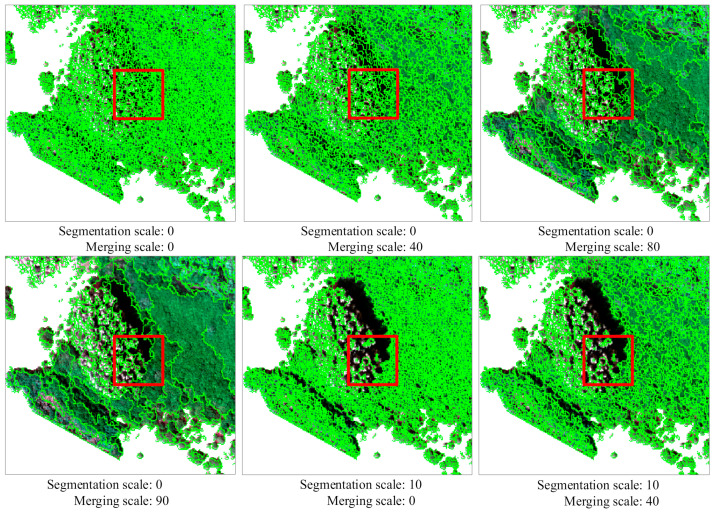
Segmentation results with different scale parameters.

**Figure 3 sensors-24-04108-f003:**
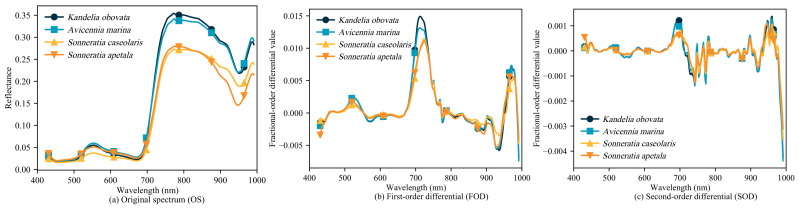
Spectral feature curves of different tree species.

**Figure 4 sensors-24-04108-f004:**
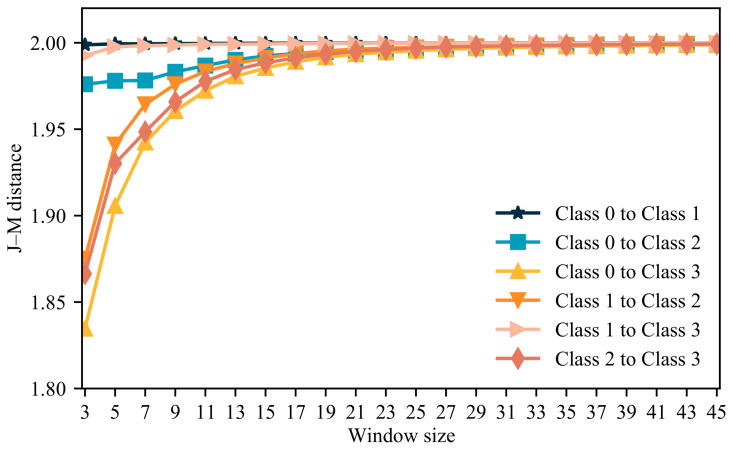
J-M distance for different windows.

**Figure 5 sensors-24-04108-f005:**
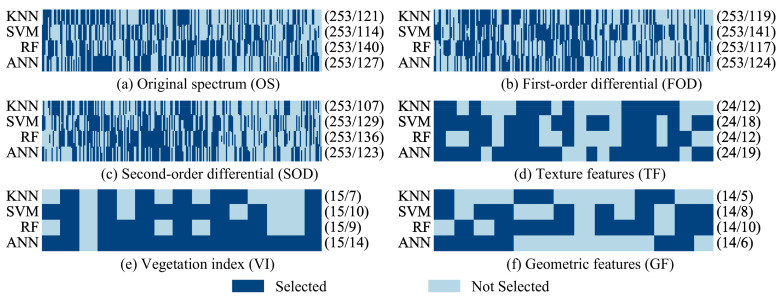
Optimal features for different methods and feature variables.

**Figure 6 sensors-24-04108-f006:**
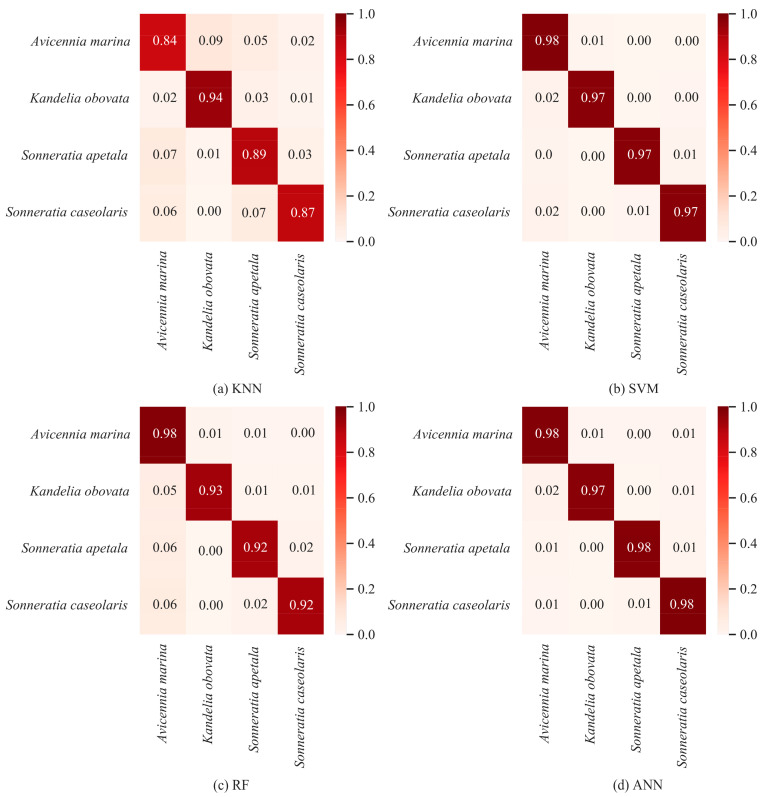
Confusion matrixes for different models.

**Figure 7 sensors-24-04108-f007:**
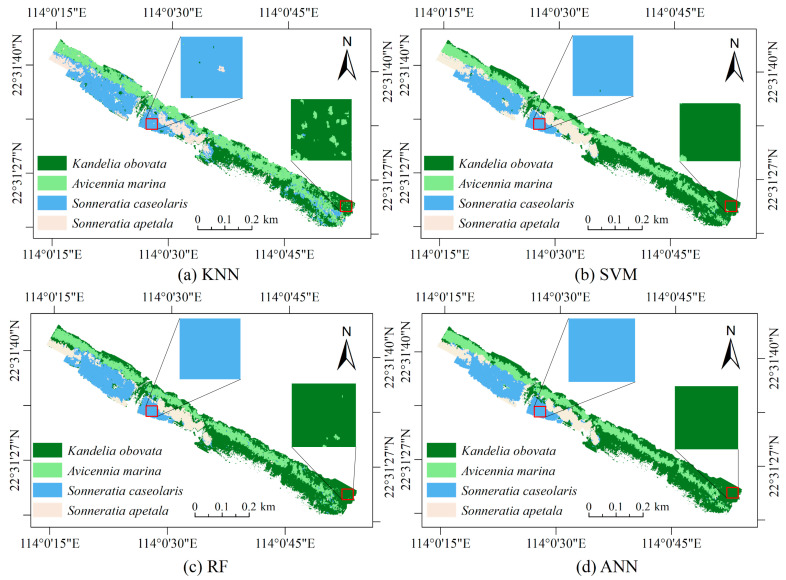
Classification results maps for different models.

**Figure 8 sensors-24-04108-f008:**
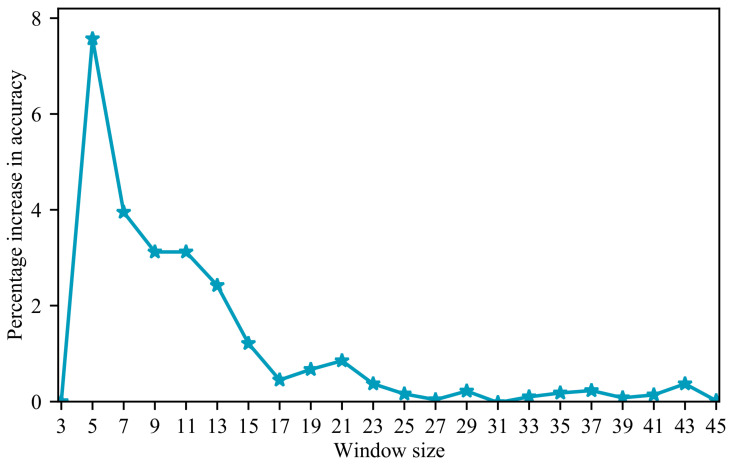
Percentage of accuracy improvement based on different window sizes.

**Figure 9 sensors-24-04108-f009:**
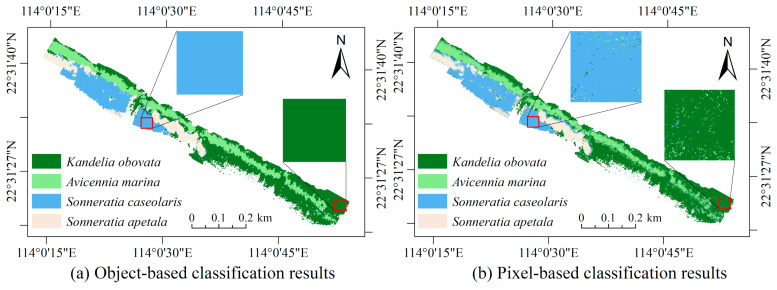
Maps of object-oriented and pixel-based classification results.

**Figure 10 sensors-24-04108-f010:**
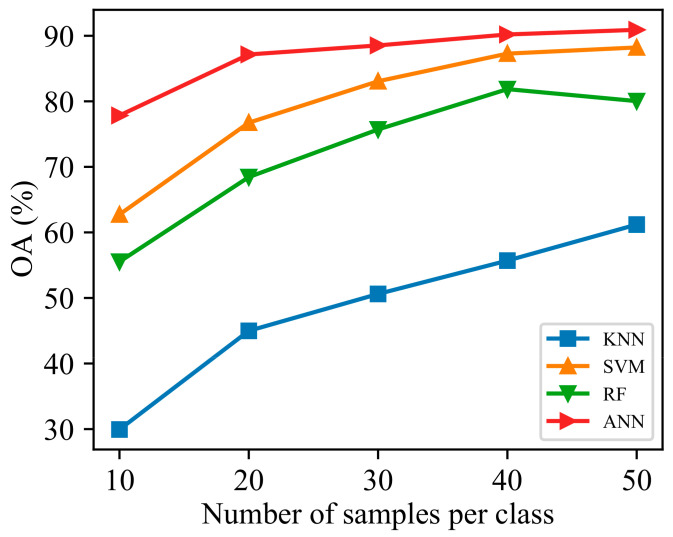
OA results for different training samples.

**Table 1 sensors-24-04108-t001:** Four mangrove species and localized views on UAV reference images.

No.	*Kandelia obovata*	*Avicennia marina*	*Sonneratia caseolaris*	*Sonneratia apetala*
RGB images	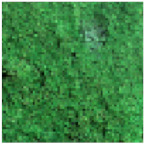	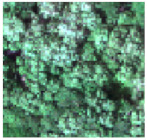	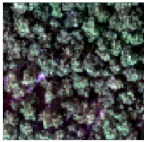	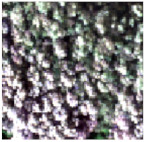
Features	Smooth color tone with no obvious textural features.	Light green in color with rougher textural features.	The color is dark with obvious textural features.	The color is bright with obvious textural features.

**Table 2 sensors-24-04108-t002:** Sample numbers for different classes.

No.	Class Name	Training	Test	Total
0	*Kandelia obovata*	795	1856	2651
1	*Avicennia marina*	390	813	1203
2	*Sonneratia caseolaris*	469	1096	1565
3	*Sonneratia apetala*	459	1167	1626
Total		2113	4932	7045

**Table 3 sensors-24-04108-t003:** Formulae for the VIs used in this study.

Vegetation Indices	Abbreviation	Formula
Normalized Difference Vegetation Index	NDVI	$\frac{\rho_{860}-\rho_{680}}{\rho_{860}+\rho_{680}}$
Blue Green Pigment Index 2	BGI2	$\frac{\rho_{450}}{\rho_{550}}$
Renormalized Difference Vegetation Index	RDVI	$\frac{\rho_{860}-\rho_{680}}{\sqrt{{(\rho }_{860}+\rho_{680})}}$
Transformed Chlorophyll Absorption in Reflectance Index	TCARI	$3\times ((\rho_{700}-\rho_{680})-0.2\times (\rho_{700}-\rho_{550})\times (\rho_{700}/\rho_{680}))$
Green Optimized Soil Adjusted Vegetation Index	GOSAVI	$\frac{\rho_{860}-\rho_{550}}{\rho_{860}+\rho_{550}+0.16}$
TCARI/GOSAVI Ratio	TCARIOSAVI	TCARI/GOSAVI
Modified Chlorophyll Absorption in Reflectance Index 2	MCARI2	$\frac{1.5\times (2.5\times (\rho_{860}-\rho_{680})-1.3\times (\rho_{860}-\rho_{550}))}{\sqrt{{\left(2\times \rho_{860}+1\right)}^{2}-\left(6\times \rho_{860}-5\times \sqrt{\rho_{680}}\right)-0.5}}$
Photochemical Reflectance Index	PRI	$\frac{\rho_{531}-\rho_{570}}{\rho_{531}+\rho_{570}}$
Transformed Difference Vegetation Index	TDVI	$\frac{1.5\times (\rho_{860}-\rho_{680})}{\sqrt{{\rho_{860}}^{2}+\rho_{680}+0.5}}$
Carotenoid Reflectance Index 2	CRI2	$\frac{1}{\rho_{510}}-\frac{1}{\rho_{700}}$
Plant Senescing Reflectance Index	PSRI	$\frac{\rho_{680}-\rho_{450}}{\rho_{750}}$
Green Leaf Index	GLI	$\frac{2\times \rho_{550}-\rho_{680}-\rho_{450}}{2\times \rho_{550}+\rho_{680}+\rho_{450}}$
Red Edge Normalized Difference Vegetation Index	RENDVI	$\frac{\rho_{750}-\rho_{700}}{\rho_{750}+\rho_{700}}$
Structure Insensitive Pigment Index	SIPI	$\frac{\rho_{860}-\rho_{445}}{\rho_{860}+\rho_{680}}$
Nonlinear Vegetation Index	NLI	$\frac{{\rho_{860}}^{2}-\rho_{680}}{{\rho_{860}}^{2}+\rho_{680}}$

**Table 4 sensors-24-04108-t004:** Comparison of classification accuracy results before and after feature variables optimization.

	No Feature Selection				Feature Selection			
Feature Variables	KNN	SVM	RF	ANN	KNN	SVM	RF	ANN
OS	71.74	95.07	80.03	91.91	74.13	95.72	80.49	91.91
FOD	85.87	95.13	91.30	95.76	88.79	95.88	92.86	96.21
SOD	86.21	93.07	92.01	95.34	90.51	94.32	92.76	96.47
TF	85.02	96.88	89.07	90.75	87.53	97.04	90.09	91.61
VI	72.51	88.93	77.01	74.53	75.04	89.03	78.61	75.18
GF	36.84	37.53	35.73	38.48	37.43	39.17	35.87	39.33

**Table 5 sensors-24-04108-t005:** Classification results for different combinations of feature variables.

No.	Feature Combinations	OA	AA	Kappa
1	OS + FOD	95.68	95.55	0.9404
2	FOD + TF	96.88	96.75	0.9569
3	OS + FOD + TF	97.38	97.26	0.9639
4	OS + SOD + TF	96.67	96.41	0.9541
5	OS + FOD + TF + VI	97.45	97.29	0.9648
6	OS + FOD + TF + GF	97.40	97.25	0.9642
7	FOD + TF + VI	97.22	97.07	0.9617
8	FOD + TF + GF	97.22	97.06	0.9617
9	FOD + TF + VI + GF	97.51	97.38	0.9656
10	OS + FOD + TF + VI + GF	97.59	97.43	0.9667

**Table 6 sensors-24-04108-t006:** Classification results of different models based on the optimal combination of feature variables. Note: the values in brackets represent the results of 10-fold cross-validation on the training set.

Class	KNN	SVM	RF	ANN
0	84.32	98.38	97.79	98.28
1	94.10	97.17	92.74	97.17
2	88.78	97.17	91.97	97.81
3	87.23	97.00	91.95	98.46
OA	87.61 (87.36)	97.59 (97.26)	94.28 (93.85)	98.03 (97.96)
AA	88.61 (88.37)	97.43 (97.17)	93.61 (93.29)	97.93 (97.98)
Kappa	0.8306 (0.8279)	0.9667 (0.9622)	0.9207 (0.9151)	0.9729 (0.9720)

**Table 7 sensors-24-04108-t007:** Classification results based on objects and pixels.

	Object-Based				Pixel-Based			
Class	KNN	SVM	RF	ANN	KNN	SVM	RF	ANN
0	84.32	98.38	97.79	98.28	82.77	97.40	96.64	97.78
1	94.10	97.17	92.74	97.17	89.46	96.65	92.81	97.49
2	88.78	97.17	91.97	97.81	88.04	95.11	91.57	95.92
3	87.23	97.00	91.95	98.46	86.42	95.53	93.95	95.97
OA	87.61	97.59	94.28	98.03	85.93	96.33	94.23	96.89
AA	88.61	97.43	93.61	97.93	86.67	96.17	93.75	96.79
Kappa	0.8306	0.9667	0.9207	0.9729	0.8078	0.9494	0.9203	0.9572

**Table 8 sensors-24-04108-t008:** Comparison of running time for different models.

Time	KNN	SVM	RF	ANN
Training time (s)	0.004	0.153	7.490	2.000
Verification time (s)	0.516	0.652	0.380	0.016

## Data Availability

The hyperspectral remote-sensing images of mangrove data were sourced from https://github.com/XUYIRS/Supplementary-file (accessed on 13 June 2023).

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
