# Peer review of "Mangrove Species Classification from Unmanned Aerial Vehicle Hyperspectral Images Using Object-Oriented Methods Based on Feature Combination and Optimization"

_sensors, 2024, doi:10.3390/s24134108_

Round 1
Reviewer 1 Report
Comments and Suggestions for Authors
1. Emphasize the highlights and contributions in your abstract.
2. Rename the Figure 1 (Figure 1 (a)xxx, (b)xxx....)
3. Typo, table 3 name.
4. Line 223, why you selected these four ?
Author Response
Thank you very much for your constructive comments on this paper. Our point-to-point response to the reviewers is attached.

Reviewer 2 Report
Comments and Suggestions for Authors
In this paper, the hyperspectral data of mangrove forests collected in Futian District, Shenzhen were used to classify mangrove species, and the results showed that the SVM classifier had the best effect, with an overall accuracy of 97.04%. The proposed method has a high classification detection rate and good preliminary test effect, but the complexity of the dataset is low, and the data analysis is relatively simple, which needs to be improved. Overall, the article demonstrates a level of innovation.
1.The dataset in this paper only uses hyperspectral images in a protected area, and please further explain whether the obtained model has strong generalization ability.
2.The dataset in this paper is collected under only one weather condition, please explain whether the model is applicable to the other weather conditions.
3.The complexity of the dataset in this paper is low, please explain whether the different planting densities of mangroves will affect the corresponding characteristics, which will affect the classification effect of the model
4.The evaluation index is relatively simple, and it is recommended to introduce the classification time to further illustrate the superiority of the model.
5. It is suggested that the confusion matrix obtained by the optimal model test should be displayed in the result analysis to improve the intuitiveness of the results.
6.In the analysis of the results, KNN performed the worst and had more misclassifications, please explain the reason.
7.In this paper, fixed parameters are used for feature extraction and classification, and please explain whether different parameters will affect the classification results.
Comments on the Quality of English Language
Minor editing of English language required.
Author Response

(The authors gave the same response as above.)

Reviewer 3 Report
Comments and Suggestions for Authors
This manuscript entitled Mangrove classification from UAV hyperspectral images using object-oriented Based on feature combination and optimization it is an interesting work however there are some major aspects that need to be clarified before publishing. This review need to be rewritten and could be published after major revisions.
Some item that need to be reviewed are:
1. Material and methods. In line 124 authors include medium and minimum temperature, they should include maximum to present all the information.
2. Line 142: “The hyperspectral image sensor covers the spectral 141 range of 400-1000 nm, with a spectral resolution of 2.8 nm, and comprises 270 spectral 142 bands”. What authors mean with bands? Will it be points that define and spectra? Please clarify.
3. Lines 310-312. Authors associate features to wavelength. For better understanding explain the importance of those wavelength and their relation with some of the features which are evaluated in this work.
4. Line 348 “Since SVM shows excellent classification performance with a single feature variable,” identify the variable with some aspect evaluated in the study (color, species, etc.)
5. In this study authors obtain and hyperspectral imaging, however to classify or to evaluate all the methods and their classification with different chemometric tools (ANN, SVM, KNN, RF). The use a spectra database as reference, but how authors have obtained or scanned that data base? Have they scanned all the samples collected? If Yes, How have they collected those spectra? Include or clarify this information in the text.
6. How do authors evaluate the accuracy? (Figure 7). Sampling has included all the scanning surface… In the paper there is no data clarifying how author have obtained reference data to assure the quality of classification when using one or other models on hyperspectral imaging.
Comments on the Quality of English Language
No comments
Author Response

(The authors gave the same response as above.)
